# Effect of Solvent and Grain Color on the Biological Activities of Maize Grain

**DOI:** 10.3390/foods14071163

**Published:** 2025-03-27

**Authors:** Yolanda Salinas-Moreno, Miguel Ángel Martínez-Ortiz, Eduardo Padilla-Camberos, José Luis Ramírez-Díaz, Alejandro Ledesma-Miramontes, Ivone Alemán de la Torre, Alberto Santillán-Fernández

**Affiliations:** 1Programa de Maíz, Campo Experimental Centro Altos de Jalisco, Instituto Nacional de Investigaciones Forestales, Agrícolas y Pecuarias (INIFAP), Tepatitlán de Morelos 47600, Jalisco, Mexico; martinez.miguel@inifap.gob.mx (M.Á.M.-O.); ramirez.joseluis@inifap.gob.mx (J.L.R.-D.); ledesma.alejandro@inifap.gob.mx (A.L.-M.); aleman.ivone@inifap.gob.mx (I.A.d.l.T.); 2Centro de Investigación y Asistencia en Tecnología y Diseño del Estado de Jalisco A.C., Guadalajara 44270, Jalisco, Mexico; epadilla@ciatej.mx; 3Colegio de Postgraduados, Campus Campeche, Sihochac, Champotón 24450, Campeche, Mexico; santillan.alberto@colpos.mx

**Keywords:** *Zea mays* L., phenolic compounds, antioxidant capacity, antimutagenic activity

## Abstract

The color of maize grain, ranging from pink to purple, is related to the presence of phenolic compounds whose efficient extraction is affected by the solvent used. This study aimed to determine the effect of solvents and maize grain color on the phenolic composition and biological activities of maize extracts. Six samples (two with brick red, BR, two with cherry red CR, and two with blue–purple BP) of maize grain were used. The solvents were acidic methanol (MTFA) and aqueous acetone (AWAC). The phenolic composition was evaluated by total soluble phenolics (TSPs), anthocyanins (TACs), flavonoids (FLAVs), and proanthocyanidins (PAs). Biological activities evaluated were antioxidant capacity (AC), antifungal activity (AFA) and antimutagenic (AM) activity. The type of solvent used exerted a higher effect than the maize grain color on the phenolic composition of biological activities. The TAC and FLAV variables were more influenced by solvent than TSPs and PAs, while AC was affected only when evaluated by the DPPH method. AWAC extracts showed AFA and had the highest AM, unlike MTFA extracts. These results highlight the importance of selecting an appropriate solvent to maximize the functional properties of maize grain extracts and reach a more objective evaluation of the potential of food on its biological activities.

## 1. Introduction

Maize grain has a high amount of phenolic compounds (PCs), integrating both soluble or extractable phenolics and insoluble or bound phenolics [1,2]. Although the soluble phenolic fraction is less abundant, it exhibits the greatest diversity in these types of compounds. In maize grain with color ranging from pink to deep purple, the color is due to the presence of anthocyanins that accumulate in the pericarp, the aleurone layer or both grain structures [1,3]. In these maize grain colors, the soluble phenolic (SP) fraction contains a wide variety of biological compounds, among which flavonoids of the anthocyanin and proanthocyanidin types, phenolic acids, flavonols, and phenolic amides are included [3,4,5]. As phenolic compounds, all these chemical compounds share the presence of at least one phenol group in their chemical structure. However, each one possesses particular polarity and extractability in certain solvents, depending on its chemical structure, which also affects its biological activities [6].

Phenolic compounds in plant matrices exhibit varying degrees of polarity, allowing them to be extracted selectively using solvents with differing polarities. Polar solvents such as methanol or acetone improve solubility, while acidic conditions (pH 1–3) improve the stability of phenolic compounds. When solvents are applied in a sequential extraction process, the results often differ from those obtained using solvents individually. Rocha-Guzmán et al. [7] demonstrated this by extracting phenolic compounds from bean cotyledons using successive treatments of 70% aqueous acetone and 50% aqueous methanol. They found that aqueous acetone yielded a higher phenolic content compared to aqueous methanol. Similarly, Mokrani and Madani [8] reported that acetone was more effective for extracting soluble phenolics from peaches, whereas 60% ethanol was the best solvent for extracting flavonoids from this plant matrix. Hapsari et al. [9] obtained a higher yield of flavonoids from dried leaves of *Calophyllum inophyllum* with methanol compared to acetone. These findings highlight the importance of selecting an appropriate solvent tailored to the specific phenolic compounds of interest and the characteristics of the plant matrix involved.

Maize grains with anthocyanins, particularly those with purple and blue tones, are the most studied for their phenolic content [10,11,12] and biological activities [10,11], with acidified methanol-based mixtures being the preferred solvent for extracting polar phenolic compounds like anthocyanins and phenolic acids. However, maize grains in shades from pink to purple also contain less-studied, low-polarity phenolic compounds such as flavan-3-ols and phlobaphenes [13,14], which are poorly extracted by polar solvents.

The majority of the published studies on the quantification and characterization of soluble phenolic compounds from pigmented maize grain have been conducted using methanol or acetone-based solvents [5,15,16,17], and fewer studies have employed two or more solvents to quantify and analyze different phenolic compound groups [10,13].

The phenolic composition of maize grains with different color tones has been partially studied [10,13,18], and there are limited reports on how the type of solvent and grain color influence biological activities. Therefore, this study aimed to determine the effect of solvents and grain color on the soluble phenolic composition, as well as the antioxidant, antifungal, and antimutagenic activities of maize grain extracts.

## 2. Materials and Methods

### 2.1. Plant Material and Extract Preparation

A set of six native Mexican samples of maize grain was used: two with brick red (BR) grain, two with cherry red (CR), and two with blue–purple (BP) grain (Table 1). These three maize grain colors are among the most common in Mexican maize diversity. Because this is an exploratory study, we used only two samples of each grain maize color.

The maize grain samples were obtained from Mexican germplasm banks. Around 30–40 grains were ground using a cyclone mill (UDY Corporation, Fort Collins, COL, USA) with 0.5 mm mesh, and the resulting flour was subjected to extraction processes. The solvents used were methanol acidified with trifluoroacetic acid at 1% (MTFA) and a mixture of acetone–water–acetic acid (AWAC) in proportions of 75:24.5:0.5 *v*/*v*/*v*. Previous research has demonstrated the efficiency of these solvents and conditions for the extraction of bioactive compounds [19]. For extract preparation, one gram of ground and defatted maize grain from each accession was extracted with 20 mL of solvent. The sample was sonicated (Branson 2150 sonication bath, Danbury, USA) for 15 min, after which it was refrigerated for 105 min. Each sample was centrifuged (Mod. Universal 32, Hettich zentrifugen, Tüttlengen, Germany) at 2200× *g* for 10 min. The supernatant was recovered and filtered using Whatman no. 4 paper, and the volume was measured. These extracts were used for phenolic compound analysis and biological activity evaluation. For preparing the extracts to be used in the antifungal and antimutagenic activity assays, 5 mL of the original extract was concentrated to dryness at reduced temperature and pressure in a rotary evaporator and re-dissolved in 5 mL of aqueous DMSO at 5 ppm. The no-cell toxicity of the DMSO solution was previously tested at the concentration used.
foods-14-01163-t001_Table 1Table 1General information on the native maize grain from the accessions used in this study.Maize Grain IdentityColor of the GrainPigment Location ^◊^Endosperm Type ^◊^Maize Race Name ^†^Code of the Accesión ^†^Main Food Uses ^†^Grain PictureBR03Brick redPericarpFlouryOlotónCHIS-1082Tortillas
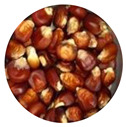
BR06Brick redPericarpFlouryDulceJAL-188Pinole
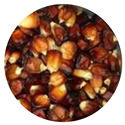
CR14Cherry redAleuroneFlouryElotes CónicosQRO-94Tortillas
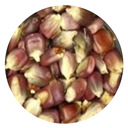
CR20Cherry redAleuroneFlouryElotes CónicosPUE-149Tortillas
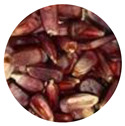
BP36Blue–purpleAleuroneFlouryBolitaOAX-766Tlayudas, tortillas
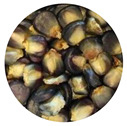
BP51Blue–purpleAleuroneFlouryElotes OccidentalesNAY-38Pozole, tortillas and immature corn
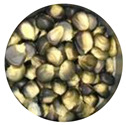
**^†^** Information obtained from the CONABIO [20] database. Pinole is roasted maize grain, ground and mixed with sugar. Tlayudas and pozole are traditional Mexican dishes. **^◊^** Own information.

### 2.2. Phenolic Compounds Analysis

Total soluble phenolic (TSP) compounds were determined using the Folin–Ciocalteu method [21]. The results were expressed as mg of ferulic acid equivalents per kg of dry weight (mg FAE/kg DW) (Appendix A).Total anthocyanins (TACs) were quantified using a spectrophotometer (Lambda 25 UV/Vis, Perkin Elmer, Waltham, MA, USA) at a wavelength of 520 nm. The results were expressed as milligrams of cyanidin-3-glucoside equivalents per kilogram of dry weight (C3GE/kg DW) (Appendix A), following the methodology described by Salinas-Moreno et al. [22].Total flavonoids (FLAVs) were quantified using the method described by Sumczynski et al. [23], and results were expressed as µg equivalents of catechin (CE)/g DW (Appendix A).Proanthocyanidins (PAs) were determined by the method of DMAC (4-(Dimethyl amino) cinnamaldehyde) described by Wallace and Giusti [24]. The results were reported in terms of µg equivalents of catechin/g of DW (Appendix A).

### 2.3. Biological Activities Analysis

#### 2.3.1. Antioxidant Capacity

ABTS method: Antioxidant capacity was determined using the 2,2′-azino-bis(3- ethylbenzothiazoline-6-sulphonic acid) (ABTS) assay, as described by Re et al. [25]. The measurements were performed in triplicate, and antioxidant capacity was expressed as μmols of Trolox equivalents per gram of dried weight (DW).

DPPH method: The estimation of the Trolox equivalent antioxidant capacity was determined using a 60 μM solution of 1,1-diphenyl-2-picryl-hydrazyl (DPPH), using the method reported by Brand-Williams et al. [26]. The measurements were performed in triplicate, and AC was expressed as μmols of Trolox equivalents per gram of DW.

FRAP (ferric ion-reducing antioxidant power assay) method. The methodology of Benzie and Strain [27] was used. Results were expressed in μmols of Trolox equivalents per gram of DW. A standard curve of this synthetic antioxidant was prepared.

#### 2.3.2. Antifungal Activity

The antifungal activity was evaluated using the plate microdilution test [28] with the incorporation of the Alamar Blue compound (ABC). ABC is a dye that changes from blue to pink upon oxidation by the growth of the microorganism. The color change was used to indirectly measure the fungal growth, which was quantified using a spectrophotometer. This test was considered a presumptive indicator of antifungal activity. A strain of the fungus *Fusarium oxysporum* provided by the National Center for Genetic Resources (CNRG) of the National Institute of Forestry, Agriculture and Livestock Research (INIFAP) was used. The identity of this strain in the CNRG is CMCNRG1007, and it is classified as virulent. From each extract obtained with the two solvents, 5 mL was evaporated in a rotary evaporator (Buchi model R-215, Labortechnick AG, Flawil, Switzerland) to eliminate the organic solvent. The volume was then adjusted to 2.0 mL with aqueous DMSO at 5 ppm.

The strain was activated, and a conidia suspension was prepared and adjusted to 2 × 10^5^ CFU/mL. For the assay, 50 µL of the conidia suspension was placed in each well of the microplate, and 50 µL of the extract to be evaluated was added. As a negative control, 50 µL of the conidia suspension was used. For the positive control, 50 µL of the commercial antifungal Itraconazole at 20 ppm was used (20 ppm was the MIC). The microplate was incubated at 30 °C for 24 h. At the end of this first incubation, 10 µL of ABC was added to each well, and the microplate was incubated for an additional 4 h at the same temperature inside a Multiskan™ GO (model 1510, Thermo Fisher Scientific Oy, Vantaa, Finland). After the second incubation, the optical density of the wells was read at 570 nm. The change from blue to pink in the wells indicated fungal growth. The methanol extracts not only failed to inhibit fungal growth but actually promoted it, so it was decided to continue only with the acetone extracts in the evaluation of the antifungal activity using the percentage inhibition test [29]. For this test, a fungal growth sampling was placed in the center of a potato dextrose agar plate, to which 500 µL of the extract to be evaluated had been added before solidification. The plates were incubated in the dark at 30 °C for the time required for the positive control (without adding extract) to completely cover the plate. For the negative control, 500 µL of Itraconazole was added at the fungus’s minimum inhibitory concentration (20 ppm). The analyses were performed in triplicate. The percentage of inhibition was calculated using the following Formula (1):(1)$$\mathrm{Percentage}\;\mathrm{of}\;\mathrm{inhibition}\;\left(\mathrm{PI}\right)=\left(\frac{\mathrm{MGPC}-\mathrm{MGT}}{\mathrm{MGPC}}\right)*100,$$
where MGPC = mycelial growth of the positive control and MGT = mycelial growth of the treatment.

#### 2.3.3. Antimutagenic Activity

The antimutagenic activity (AM) of the maize grain extracts was determined by the Ames assay [30]. The assay was applied to only one sample per color because of the exploratory scope of this study. The extracts (MTFA and AWAC) were tested at the original concentration and diluted at 50% with aqueous DMSO at 5 ppm. The genetically modified strains of *Salmonella typhimurium* TA100 and *Salmonella typhimurium* TA98 were used. After activating the strains and confirming their genotypic characteristics, they were grown in sterile broth to obtain the bacterial cell suspensions for running the assay. Methylmethanesulfonate was used as a mutagenic agent at concentrations of 1250 µg/plate for TA98 and 2500 µg/plate for TA100. A volume of 50 µL of extract was added to each plate. For the negative control, the strain was cultured without the mutagen so that the colonies that developed were considered “Revertants”. For the positive control, the mutagen was added so that the strain would lose its mutation and be able to grow in a biotin/histidine-restrictive environment. After the incubation period, the number of colonies was counted. The arithmetic average of each treatment and strain was calculated. The antimutagenic activity was expressed as the percentage of inhibition of the mutagenic activity and was calculated as indicated by Socaci et al. [31] using the following expression (2):(2)$$\%\;\mathrm{IN}\;=\left(\frac{1-T}{M}\right)*100,$$
where T = number of revertants per plate in the presence of the mutagen and the extract and M = number of revertants per plate in the positive control (without extract).

### 2.4. Statistical Analysis

The results on phenolic composition and antioxidant capacity were subjected to ANOVA analysis and mean comparisons using Tukey’s test (*p* = 0.05). Those results on antifungal and antimutagenic activities were reported as means ± standard deviation of three replicated analyses.

## 3. Results and Discussion

### 3.1. Effect of Type of Solvent in the Phenolic Composition of Maize Grain Extracts

The type of solvent used significantly affected (*p* ≤ 0.05) the amount of phenolic compounds recovered from maize flours with different grain colors. The most affected variables were TACs and FLAVs (Figure 1). For TSP, the average content was similar between the two solvents tested. This result suggests that if the objective is to extract TSPs from maize grain, either methanol (MTFA) or acetone (AWAC) can be used. Our findings are consistent with those of Socaci et al. [31], who reported similar phenolic compound levels using methanol and aqueous acetone to extract phenolic compounds from brewer’s spent grain. Additionally, no substantial differences in TSP content were observed among the maize samples.

The MTFA solvent extracted, on average, 83% more anthocyanins than AWAC. This result may be related to the pH of the solvents (MTFA: 1.1, AWAC: 3.02), as pH influences the chemical structure of anthocyanins in solution [32]. At very low pH, the flavylium cation—the most stable chemical structure of anthocyanins—predominates and absorbs strongly at the wavelength used for anthocyanin quantification (530 nm), a situation that does not occur at pH 3 or higher. However, differences in the polarity of the solvents could also contribute to variations in phenolic compound recovery and stability. In acidified acetone solvent, anthocyanins from strawberry purees degraded within 48 h at 4 °C [33].

For TACs, the BP maize grain samples exhibited the highest values, while the BR maize samples showed the lowest. Among the maize grains containing anthocyanins, BP grains had higher anthocyanin levels than CR grains [5]. In contrast, the anthocyanin content in BR maize grains was minimal, as the pericarp brick red coloration of these grains is due to phlobaphenes [34,35], which are phenolic pigments distinct from anthocyanins.

The AWAC solvent favored the extraction of FLAVs, yielding approximately 70% more than MTFA. This result may be attributed to the polarity of the solvents; acetone, being less polar than methanol, facilitates the extraction of phenolic compounds with lower polarity, such as dimers and oligomers of catechin/epicatechin and other flavanols [13]. Among the maize grains, the CR variety exhibited higher FLAV content compared to the BP and BR varieties.

Socaci et al. [31] reported that 60% acetone was the best solvent for extracting flavonoids from brewer’s spent grain. However, the efficiency of extraction using aqueous acetone is affected by the type of flavonoids. Using the percolation method, a higher yield of flavonoids from dried leaves of *Calophyllum inophyllum* with methanol compared to acetone was obtained by Hapsari et al. [9]. Knowing the chemical characteristics of the phenolic compounds is essential for selecting the most suitable solvent for their extraction.

Among the flavonoids, other than anthocyanins described in maize grains, quercetin and kaempferol derivatives are the most common [11,36], along with some catechin oligomers. These compounds are generally less polar than anthocyanins or phenolic acids, making them more effectively extracted with acetone-based solvents.

The type of solvent had a minimal impact on the recovery of PAs, with slightly higher recovery observed using MTFA compared to AWAC. Among the maize samples, CR20 exhibited the highest PA content with both solvents. The content of PAs (also known as condensed tannins) in maize grain or co-products from its transformation through processing as dry or wet milling was analyzed by Chen et al. [13] in purple, red, and blue maize grains using acidic methanol as solvent. The authors reported the highest content of PAs in purple maize, followed by red and blue maize. The PA values they reported were much higher than the values we obtained, probably due to differences in the quantification method used. They used the vanillin method, while we used the DMAC method. With the vanillin method, the presence of anthocyanins causes interference because the wavelength used (510 nm) to quantify the reaction between PAs and the reagent falls within the range at which anthocyanins absorb [37]. The study of PAs in maize grains has been limited, likely due to their low concentrations in this cereal and the complexity of their structure, which makes analysis challenging.

### 3.2. Antioxidant Capacity

The antioxidant capacity (AC) of maize grain samples extracted with two solvents and evaluated using the ABTS, DPPH, and FRAP methods is presented in Figure 2. Among the methods, ABTS yielded the highest antioxidant capacity values (10.65–23.93 µmol TE/g DW), likely due to the high sensitivity of the ABTS radical to phenolic compounds in maize grains [17,38]. On average, extracts obtained with MTFA showed higher antioxidant capacity than those with AWAC, possibly because of the higher total anthocyanin content (TAC) in these extracts, particularly in CR and BP maize grains, which are rich in anthocyanins. Anthocyanins, among the phenolic compounds found in pigmented maize grains, are known to exhibit the highest antioxidant capacity [10].

Interestingly, BR grain samples exhibited the highest antioxidant capacity when extracted with AWAC (Figure 2A) despite having the lowest TAC values (Figure 1). This suggests that their high antioxidant capacity may be attributed to phenolic compounds of lower polarity, among which are some flavanols like catechin and epicatechin or condensed forms of these compounds [39]. The characteristic color of BR maize grain is mainly attributed to the presence of phlobaphenes [14], which are phenolic compounds quantified together with PAs and whose antioxidant potential has been poorly explored.

In contrast, CR maize grains showed the lowest antioxidant capacity among the samples extracted with AWAC. The color of these maize grains is mainly due to pelargonidin derivatives [12], which are less potent antioxidants than cyanidin derivatives predominant in BP maize grains [40].

Using the DPPH method, the antioxidant capacity of maize extracts varied between 0.56 and 7.25 µmols TE/g DW. The highest values were obtained in the MTFA extracts. With this solvent, the best antioxidant capacity was in the CR20 sample. However, in the AWAC solvent, the two samples with BR grain color again showed the highest antioxidant capacity. These results suggest that the antioxidant capacity in the BR maize samples is likely due to the presence of low-polarity phenolic compounds, such as phlobaphenes, which are responsible for the BR color of these maize samples [14,41].

The FRAP method yielded AC values ranging from 3.99 to 7.69 µmols TE/g DW. No significant differences (*p* > 0.05) were observed in the antioxidant capacity of the maize extracts obtained with the two solvents or among the three grain colors.

### 3.3. Antifungal Activity

We presented results of antifungal activity (AFA) for AWAC extract only because the absorbance values obtained with the MTFA extracts in the microdilution test were all close to 0.189, which is the absorbance of the negative control. This result indicates that, at the concentrations used, the MTFA extracts did not affect the growth of the fungus.

The results for the antifungal activity of AWAC extracts are shown in Table 2. The absorbance values were lower than those of the positive control for some extracts (samples CR20, BR03, and BR06). According to the microdilution test, if the extracts inhibit or control fungal growth, the blue color of the ABC, added as an indicator of growth, would remain unchanged. On the contrary, if the extracts do not inhibit fungal growth, the ABC is reduced, and absorbance values become lower than those of the negative control because of the change in color from blue to pink. Figure 3 shows the fungal growth plates for the most effective samples in terms of fungal growth inhibition, including the positive and negative controls.

The percentage of fungal inhibition observed in the extracts from maize grains ranged from 16.5 to 51.5%. None of the extracts reached the level of fungal inhibition achieved by Itraconazole at 20 ppm (positive control); however, the extract from sample CR20 came closest to this value. This maize sample showed the highest FLAV content when extracted with the AWAC solvent and the highest PA content with the two solvents (Figure 1).

The extracts evaluated for their biological activities are the soluble phenolic compound fraction, whose composition could vary with the solvent used for extracting the phenolic compounds. In general, the soluble fraction of phenolic compounds from pigmented maize grains is higher than that from white maize grains [3], mainly due to the presence of anthocyanins. In maize grains, acetone facilitates the extraction of phenolic compounds with lower polarity, such as catechin, epicatechin, and some dimers or oligomers [13,36] that could have antifungal activity. However, the concentrations of these compounds in maize grain are likely low, which may explain the weak fungal inhibition observed in this study.

In vegetal matrices different to maize grains, the phenolic compounds reported to have antifungal activity are the proanthocyanidins from grape seeds, which have shown values of percentage of fungus inhibition of 12–15% against strains of *Aspergillus niger* [42]. Similarly, ellagitannins (punicalagins) from the peel of *Punica granatum* are effective in controlling the growth rate of *Fusarium* spp. in a dose-dependent manner [43].

### 3.4. Antimutagenic Activity

The extracts of AWAC showed cytotoxicity at the original concentration because no microbial growth was observed, which suggests the presence of compounds at a concentration enough to exert antimicrobial activity. The MTFA extracts exhibited a slightly lower cytotoxic effect than the AWAC extract while permitting the growth of several colonies with each grain extract (14–36 UFC per plate). The cytotoxicity observed in the AWAC extract of maize grain samples is possibly attributed to the composition of phenolic compounds with lower polarity, influenced by each type of solvent and the concentration of the different phenolic variables in the extracts before they were concentrated to dryness for use in the evaluation of antimutagenic activity (Table 3). Of these variables, the FLAV content in the AWAC extract was between two and four times higher than that in MTFA. Among the flavonoids present in maize grain with potential antimicrobial activity, the subgroup of flavanols includes catechin, epicatechin, proanthocyanidins, and some quercetin derivatives [44]. Therefore, this group of phenolic compounds could be associated with the observed cytotoxicity.

The diluted AWAC extract showed different inhibition values with each strain. For TA98, higher inhibition values were obtained (72.7 to 85.6%) than for TA100 (15.9 to 40.2%). The differences are attributed to the particular genetic modification each strain has.

For the TA100 strain, differences among the maize samples analyzed were observed (Table 4). The highest percentage of inhibition was recorded in the BP maize sample, and the lowest was observed in RC. The values of the undiluted MTFA extract and TA98 strain were 89.4, 81.1, and 72.3% for BP, RC, and BR samples, respectively. For the TA100 strain, the values were lower (43.9, 38.9, and 41.4% for BP, RC, and BR, respectively). When diluting the extract, the percentage inhibition values for TA98 decreased slightly, whereas for TA100, they increased. This may be due to the TA100 strain being more sensitive to the presence of phenolic compounds. Socaci et al. [31] reported that the antimutagenic activity of spent brewery grain extracts was higher in samples with lower total soluble phenol content and antioxidant activity. In contrast, Pedreschi and Cisneros-Zevallos [10] found a positive relationship between antimutagenic activity and antioxidant activity when evaluating different fractions of phenolic compounds obtained from purple maize grain. The authors reported a direct relationship between the amount of phenolic compounds, expressed in µg equivalents of chlorogenic acid/plate, and the percentage of inhibition of mutagenicity towards TA98.

Mendoza-Díaz et al. [45] evaluated the antimutagenic activity of extracts from blue, red, yellow, and white grains. For blue maize, they observed inhibition percentages of 56 and 62% for strains TA98 and TA100, respectively. In the red maize, the values were 53 and 57% for the strains, in the same order. The yellow and white maize grain extracts showed lower antimutagenic potential than those observed in the maize grains with anthocyanins. According to Socaci et al. [31], a compound or extract is considered to have high antimutagenic activity when the inhibition percentage is ≥40%, intermediate when the values are between 40% and 25%, and low or null when the inhibition percentage is less than 25%.

An objective comparison of results is difficult because the concentration of phenolic compounds differs across studies. Although the Ames method is used in most of them, each study applies particularities that prevent an objective comparison. However, by the results obtained in the present study and those reported by others [10,45], it seems that anthocyanins in maize grains are associated with the antimutagenic activity exhibited by this cereal in vitro.

## 4. Conclusions

The type of solvent used to extract phenolic compounds from maize grain significantly influenced the antioxidant, antifungal and antimutagenic activities of the extracts, whereas the grain color had a lesser effect. The impact of the solvent on antioxidant capacity was evident when evaluated using the DPPH method, although this effect was less clear with other methods. Methanol extracts, which were rich in anthocyanins, showed no inhibitory effect on the growth of the tested fungal strain. In contrast, acetone extracts exhibited some antifungal activity, although neither extract performed as effectively as the commercial antifungal product. Additionally, the antimutagenic activity of the extracts was influenced by the solvent type, with the methanol extract demonstrating the highest activity due to its ability to extract anthocyanins. These findings highlight the importance of selecting an appropriate solvent to maximize the functional properties of colored maize grain extracts and to obtain a more objective evaluation of them, while the color of the maize grains studied showed a modest role in determining the biological activities. Future studies should focus on evaluating how the type of solvent influences other biological activities of maize grains, exploring a higher number of samples, and analyzing the main phenolic compounds in each solvent–maize grain extract.

## Figures and Tables

**Figure 1 foods-14-01163-f001:**
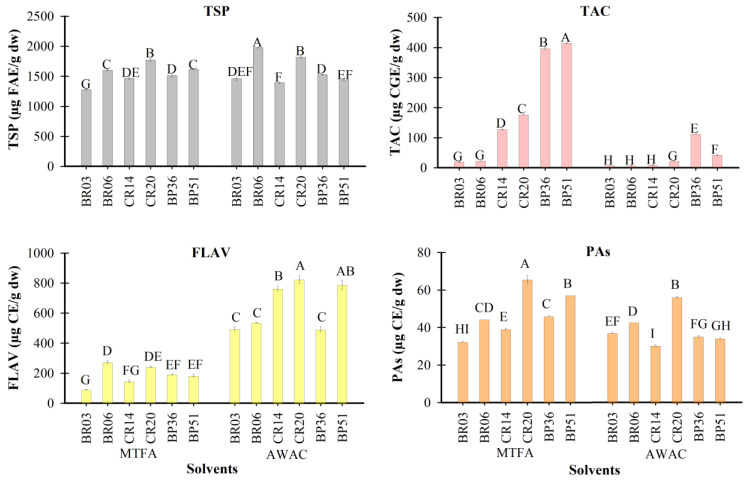
Total soluble phenolics (TSPs), total anthocyanins (TACs), flavonoids (FLAVs), and proanthocyanidins (PAs) in the blue–purple (BP), cherry red (CR), and brick red (BR) maize grain extracts obtained with methanol (MTFA) and acetone (AWAC). Bars with the similar letters mean no statistical differences (Tukey, *p* ≤ 0.05).

**Figure 2 foods-14-01163-f002:**
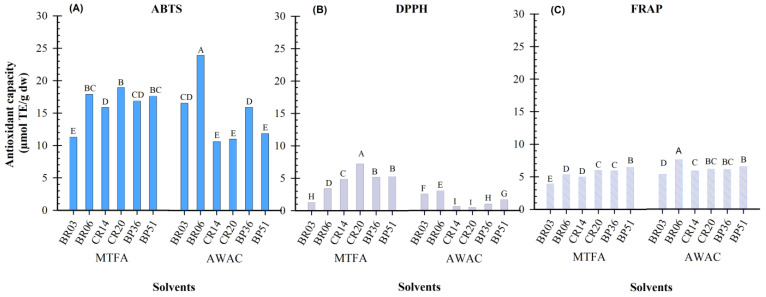
Antioxidant capacity of extracts from maize grains of different colors, obtained using two solvents and determined using (**A**) ABTS, (**B**) DPPH, and (**C**) FRAP methods. Bars with the similar letters mean no statistical differences (Tukey, *p* ≤ 0.05).

**Figure 3 foods-14-01163-f003:**
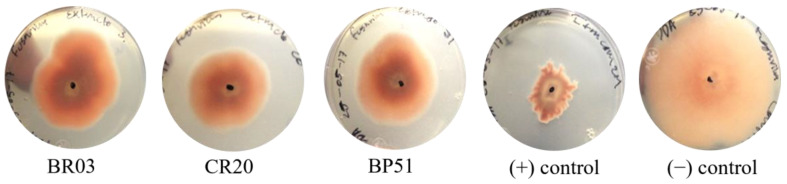
Plates showing the inhibition growth of *Fusarium oxysporum* added with the AWAC solvent extracts from maize samples with brick red (BR03), cherry red (CR20), and blue–purple (BP51) grain colors.

**Table 2 foods-14-01163-t002:** Absorbance values from the microdilution test and percentage of inhibition of *Fusarium oxysporum* growth by extracts from maize grains of different colors, obtained using the acetone-based solvent (AWAC).

Grain Color	Sample ID	Acetonic Extract (AWAC)	
		OD_570nm_	PFI
BR			
	03	0.133 ± 0.001	41.5 ± 0.3
	06	0.132 ± 0.001	21.0 ± 0.4
CR			
	14	0.134 ± 0.001	19.0 ± 0.1
	20	0.130 ± 0.002	51.5 ± 0.2
BP			
	36	0.136 ± 0.001	16.5 ± 0.2
	51	0.137 ± 0.002	43.5 ± 0.1
Positive control (Itraconazole)		0.135 ± 0.001	63.0 ± 0.2
Negative control		0.189 ± 0.001	0.0

BR: brick red grain, CR: cherry red grain, and BP blue–purple grain.

**Table 3 foods-14-01163-t003:** Concentration (µg/mL) of different phenolic variables in extracts obtained with two solvents from different maize grain colors used in the antimutagenic activity evaluation.

Extract	TAC	TSP	FLAV	PAs
AWACBR06	0.37 ± 0.01	105.39 ± 0.49	28.64 ± 0.06	2.29 ± 0.0
AWACCR20	1.19 ± 0.00	97.09 ± 1.23	44.0 ± 1.41	3.01 ± 0.02
AWACBP51	2.35 ± 0.01	78.93 ± 0.09	43.32 ± 1.67	1.87 ± 0.02
MTFABR06	1.23 ± 0.0	85.54 ± 0.7	14.64 ± 0.58	2.37 ± 0.0
MTFACR20	3.56 ± 0.02	71.57 ± 1.1	9.80 ± 0.22	2.71 ± 0.08
MTFABP51	11.06 ± 0.03	86.17 ± 0.06	9.84 ± 0.48	3.06 ± 0.0

AWAC: acetone–water–acetic acid; MEFA: methanol acidified at 1% with TFA.

**Table 4 foods-14-01163-t004:** Antimutagenic activity of extracts from pigmented maize grains, obtained using two solvents and evaluated using *S. typhimorium* strains TA98 and TA100.

Type of Solvent	Sample	Number of Revertants			
		TA98		TA100	
		Mean	% IN	Mean	% IN
	Control (−)	32 ± 0.6		90.5 ± 3.5	
AWAC	BP51	NG	------	NG	------
	CR20	NG	------	NG	------
	BR06	NG	------	NG	------
AWAC †					
	BP51	36 ± 4.6	72.7	237 ± 5.7	40.2
	CR20	19 ± 2.7	85.6	330 ± 1.73	16.7
	BR06	NG		325 ± 3.0	17.9
MTFA					
	BP51	14	89.4	222	43.9
	CR20	25	81.1	242	38.9
	BR06	36.5	72.3	232	41.4
MTFA †					
	BP51	29.3 ± 2.5	77.6	131 ± 3.0	66.9
	CR20	36.5 ± 5.5	72.3	152 ± 10.3	61.6
	BR06	23 ± 3.0	82.6	168 ± 4.6	57.6
	MMS ^a^	132 ± 7.4		396 ± 22.7	

^a^ MMS: methyl methanesulfonate (positive control); NG: no growth of colonies; AWAC: acetone–water–acetic acid; MEOH: methanol acidified at 1% with TFA; †: the extract was diluted at 50%.

## Data Availability

The original contributions presented in the study are included in the article/Appendix A, further inquiries can be directed to the corresponding author.

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
