# Peer review of "Effect of Solvent and Grain Color on the Biological Activities of Maize Grain"

_foods, 2025, doi:10.3390/foods14071163_

Round 1

Reviewer 1 Report

Comments and Suggestions for Authors

Abstract: According to the Foods template the abstract should be a total of about 200 words maximum. Please,  adjust the extension. 

Line 38: It is suggested to add another citation, in addition to this one, that does not belong to the authors of this work.

Line 66: It is suggested revising these. It is proposed: type of extract.

Line 72 to73: Justify why only use two samples per color. Statistical analysis with n= 2.

Line 80 to 82: A justification for the choice of solvents is needed for clarity.

Line 92 to 105: IIt is suggested that the authors should present as supplementary data at least the specificity and the accuracy of the methods used for quantification.

Line 124: Replace the comma with a period.

Line 139: Add meaning of AA abbreviation and include it in Abbreviations item. 

Line 158: Justify why the number of samples was reduced to one for this assay.

Line 196 to 198: It is suggested to modify this phrase. Quantification is only a measure of the extraction of the compounds and not a justification of the amount of compounds extracted.

Line 198 to 199: Polarity and pH influence the compounds that are extracted. It is suggested that the authors describe it as a statement.

Line 202 to 204: Add citations.

Line 271 to 280: Methanol is a solvent with chemical characteristics that make it very non-selective. Have the authors considered this?

Line 280: correct to phlobaphenes.

Line 316 to 317: Discuss what this differential activity may be due to.

Line 340 to 342: Discuss the reason for the antifungal activity in corn kernels.

Line 359 to 360: Add citations that justify the cytotoxicity of flavonoids.

Line 392 to 394: Explain what are the differences for which the results cannot be compared or delete this phrase

Line 400 to 404: I suggest reviewing the wording of this paragraph. The solvent influences the type of metabolites extracted from the corn kernel, which determines the observed activities, as the authors conclude in the text.

Funding information: Add funding information according to Foods template:“This research received no external funding” or “This research was funded by NAME OF FUNDER, grant number XXX” and “The APC was funded by XXX”.

Abbreviations: For better clarity and ease of understanding, it would be helpful to limit the use of acronyms in the text as too many may hinder comprehension.

Author Response

Comments 1: Abstract: According to the Foods template the abstract should be a total of about 200 words maximum. Please, adjust the extension.

Response 1:

Thank you for your comment. The requested modification has been made.

[lines 16-31]

Comments 2: Line 38: It is suggested to add another citation, in addition to this one, that does not belong to the authors of this work.

Response 2:

The requested modification was done.

[line 37]

Comments 3:  Line 66: It is suggested revising these. It is proposed: type of extract.

Response 3:

Thank you for your comment. We modified the sentence with your suggestion.

[line 69]

Comments 4: Line 72 to73: Justify why only use two samples per color. Statistical analysis with n= 2.

Response 4:

The requested modification has been made.

[lines 73-79]

Comments 5: Line 80 to 82: A justification for the choice of solvents is needed for clarity.

Response 5:

Thank you for your comment. The requested modification has been made.

[lines 84-85]

Comments 6: Line 92 to 105: It is suggested that the authors should present as supplementary data at least the specificity and the accuracy of the methods used for quantification.

Response 6:

We added additional information.

[Supplementary Materials]—calibration curves for phenolic compounds analysis (Figure S1-S4)

Comments 7: Line 124: Replace the comma with a period.

Response 7:

We changed the comma with a period.

[ line 134]

Comments 8: Line 139: Add meaning of AA abbreviation and include it in Abbreviations item.

Response 8:

We have modified “AA” by ABC and include it in the Abbreviations list.

[lines 133, 148, 329]

Comments 9: Line 158: Justify why the number of samples was reduced to one for this assay.

Response 9:

Thank you for your comment. The development of this assay was complex, and working with a larger number of samples was challenging due to time and resource constraints. We added some text in line 167,

Comments 10: Line 196 to 198: It is suggested to modify this phrase. Quantification is only a measure of the extraction of the compounds and not a justification of the amount of compounds extracted.

Response 10:

We modified the redaction to improve clarity and added some references.

[lines 205-206]

Comments 11: Line 198 to 199: Polarity and pH influence the compounds that are extracted. It is suggested that the authors describe it as a statement.

Response 11:

We have modified and add more information.

[line 208-210]

Comments 12: Line 202 to 204: Add citations.

Response 12:

The requested modification has been made.

[line 213-215]

Comments 13: Line 271 to 280: Methanol is a solvent with chemical characteristics that make it very non-selective. Have the authors considered this?

Response 13:

Thank you for your comment. Methanol is the most common solvent used in studies about anthocyanins in maize grain.  For this reason, we consider to incorporate this solvent in our study

Comments 14: Line 280: correct to phlobaphenes.

Response 14:

We have modified for correct the word “phlobaphenes”

[line 287]

Comments 15: Line 316 to 317: Discuss what this differential activity may be due to.

Response 15:

We improved the discussion and added more information about the requested.

 [line 348-352]

Comments 16: Line 340 to 342: Discuss the reason for the antifungal activity in corn kernels.

Response 16:  

We have added more information about the requested.

 [line 357-361]

Comments 17: Line 359 to 360: Add citations that justify the cytotoxicity of flavonoids.

Response 17:

We added one reference (line 386)

Comments 18: Line 392 to 394: Explain what are the differences for which the results cannot be compared or delete this phrase.

Response 18:

The type of mutagenic agent, what change completely the results.

[line 424]

Comments 19: Line 400 to 404: I suggest reviewing the wording of this paragraph. The solvent influences the type of metabolites extracted from the corn kernel, which determines the observed activities, as the authors conclude in the text.

Response 19:

Thank you for your comment. We have modified the redaction of the paragraph and added possible research directions.

[line 429-444]

Comments 20: Funding information: Add funding information according to Foods template: “This research received no external funding” or “This research was funded by NAME OF FUNDER, grant number XXX” and “The APC was funded by XXX”.

Response 20:

We have added: “Funding information”.

[line 457]

Comments 21: Abbreviations: For better clarity and ease of understanding, it would be helpful to limit the use of acronyms in the text as too many may hinder comprehension.

Response 21:

Thank you for your comment. We reduced the use of abbreviations in the text.

4. Response to Comments on the Quality of English Language

Point 1: The English is fine and does not require any improvement.

Response 1: We thank you this comment.

5. Additional clarifications

Reviewer 2 Report

Comments and Suggestions for Authors

The work evaluates the effect of grain color and solvent used on the properties of maize grain extracts. In general, the work is scientifically relevant but presents results from simple methodologies that can be enriched with other more modern methodologies. 

My main considerations are listed below. 

1. Abstract and introduction. It's well written.

2. Methodology. lines 88-89. It mentions that the extracts were used for biological activity analysis, but does not describe whether the extract was used with the solvent or whether the solvent was evaporated. If the extracts were not evaporated, how were the antifungal and mutagenic activity analyses carried out? Considering that one extract is methanolic and the other acetone, both are toxic to living organisms.

Also in the methodology, the authors use total compound analysis methodologies (content estimation) which have now been replaced by more sensitive methodologies such as HPLC analysis. I suggest a more complete characterization of the extracts using these HPLC-DAD or HPLC-MS methodologies. In addition, we know that corn contains carotenoids, but this class of phenolic compounds has not been evaluated. 

The biological activity tests were carried out on fungi and salmonella. I am very concerned about the question of whether or not the solvent was extracted, since the authors report that the best results were obtained with extracts using acetone as the solvent. 

The results are well presented and described and the discussion is well written. In general, the work is well written and of good quality. However, the methodologies used are weak and could be improved. 

Author Response

Thank you for your comment. The requested modification has been made.

Reviewer 3 Report

Comments and Suggestions for Authors

These findings highlight the importance of selecting an appropriate solvent for extracting phenolic compounds of maize grains with different color tones. This study also investigated the antioxidant, antifungal, and antimutagenic activities of maize grain extractions by two different solvents. The research is well done and presented. The main objections are below:

  • In the introduction and Material sections, there is no information about pigmented maize grain used in this study. How is pigmentation achieved? Put some table related to the main similarities and dissimilarities between maize grains with different color tones.  
  • What are these varieties/sorts dominantly used for? Is there any genetic modification, etc.?
  • The literature review in the Introduction section is poor.
  • It is tough to follow text because of a lot of abbreviations.
  • There is no evidence of different activity of the solvent itself. In some cases, solvents can also interact and participate in the tests.
  • The results were mostly descriptively commented. Deep understanding and explanation completely missing. Please provide satisfactory explanations. For example, what happened in BR06 in acetone regarding ABTS, FRAP, and even DPPH test, etc.
  • Antioxidants in ethanol and isopropanol reacted with DPPH less efficiently compared to those in methanol. You have a similar situation regarding acetone.
  • The discussion is also missing regarding no growth of colonies in Table 3. Why is there no growth in acetone?
  • Please provide consistency related to the sample name! Sample AZM051 is? … RC020?
  • Revised conclusions, give possible research directions, and explain the statements made.

Author Response

(The authors gave the same response as above.)

Round 2

Reviewer 1 Report

Comments and Suggestions for Authors

Line 76 to 78: This is not a valid justification. If N=2, the authors should say that it is an exploratory study.

Line 86: It is suggested to justify with scientific publications on corn grains and not only on proanthocyanidin.

Line167: This is not an adequate justification. It is suggested that the authors include a sentence referring to the scope of the study. They should clarify from the beginning that this is an exploratory study because the n=2.

Reviewer 2 Report

Comments and Suggestions for Authors

The authors added essential information that was missing from the methodology and improved other points of the work. However, I still think that an analysis of the profile of compounds present in the extracts is vital to justify the effects observed in the in vitro biological tests. 

Reviewer 3 Report

Comments and Suggestions for Authors

The manuscript is slightly improved. I recommend publications.  
